# Preferences for long-acting injectable HIV pre-exposure prophylaxis service delivery among male and female sex workers in Uganda: A discrete choice experiment

Richard Muhindo[1]*, Rachel King[2], Andrew Mujugira[3], Whitney Irie[4], Patience Muwanguzi[1], Flavia Dhikusooka[3], Joseph Musaazi[3], Barbara Castelnuovo[3]

1 Department of Nursing, College of Health Sciences, Makerere University, Kampala, Uganda,
2 Department of Epidemiology and Biostatistics, University of California, San Francisco, California, United States of America, 3 The Infectious Diseases Institute, College of Health Sciences, Makerere University, Kampala, Uganda, 4 School of Social Work, Boston College, Chestnut Hill, Massachusetts, United States of America

☯ These authors also contributed equally to this work.
* r.muhindo@yahoo.com

## Abstract

In Sub-Saharan Africa, limited data exist on the delivery of injectable HIV pre-exposure prophylaxis (PrEP). We explored service delivery preferences for injectable cabotegravir (CAB-LA) among heterosexual male sex workers (MSWs) and female sex workers (FSWs) in Uganda. We conducted a discrete choice experiment (DCE) among HIV-negative sex workers in two high HIV-prevalence Ugandan cities between October and December 2024. Participants selected from alternatives varying by location, waiting time, provider gender, and additional services. A mixed logit model identified the most valued attributes influencing preferences for CAB-LA delivery. We enrolled 251 sex workers (SWs), comprising 52 (20.7%) MSWs. MSWs were more likely to have higher education (26.9% vs 4.5%), while FSWs had been in the industry longer (24 months vs 18 months). PrEP awareness was lower among MSWs than FSWs (86.5% vs. 95.5%, p = 0.027). Overall, 39.4% reported current PrEP use. Injectable PrEP was the most preferred formulation (77.3%), and willingness to use PrEP among non-users was higher in MSWs (67.3% vs. 45.7%, p = 0.001). Regarding service delivery preferences, MSWs and FSWs both prioritized dispensing location (relative importance 81.1% MSWs, 82.9% FSWs) and provision of additional services (10.8% MSWs, 9% FSWs). In contrast, clinic waiting time and provider gender were considered the least important, with both groups assigning them a relative importance of 4%. Nevertheless, the three top-ranked CAB-LA delivery models prioritized access through private pharmacies or clinics with short waiting times, female or peer providers, and integrated health services. The preferred extra services included psychosocial support, cancer screening, and risk reduction counseling. Expanding

**Data availability statement:** Supporting information file 2: Data set analyzed in the study.

**Funding:** This work was supported by the Fogarty International Center of the National Institutes of Health (D43TW009343 to Craig Cohen). Craig Cohen is the Principal Investigator of the Global Health Training (GloCal) program at the University of California Global Health Institute (UCGHI). RM was supported as a Fogarty GloCal Fellow by UCGHI. The funders had no role in study design, data collection and analysis, decision to publish, or preparation of the manuscript.

**Competing interests:** The authors have declared that no competing interests exist.

injectable PrEP through private sector channels may improve access among SWs. Still, implementation research is needed to guide integration of supportive services and SWs' willingness to pay or co-pay for pharmacy or private clinic-based delivery.

## Introduction

Globally, sex workers (SWs) bear a disproportionate burden of HIV, accounting for 7.7% of new infections between 2010 and 2022, despite comprising only a small fraction of the general adult population [1]. In Sub-Saharan Africa (SSA), the epicenter of the HIV epidemic, SWs contribute an estimated 5.2%-15% of new infections in 2022 [1].In Uganda, HIV prevalence among female sex workers (FSWs) is particularly high, estimated at 31% [1,2]. In contrast, the epidemiology of HIV among heterosexual male sex workers (MSWs) in SSA, including Uganda, remains poorly understood, with limited data suggesting inconsistent condom use and low HIV testing uptake among MSWs [3].

Pre-exposure prophylaxis (PrEP) has been shown in clinical trials to substantially reduce the risk of HIV acquisition, offering SWs an additional effective prevention strategy [4,5]. The World Health Organization (WHO) recommended oral tenofovir disoproxil (TDF)-based PrEP in 2015, and Uganda adopted it as part of its HIV prevention strategy in 2017 [6,7]. Although oral PrEP services are available in Uganda for high-risk individuals, including sex workers [7], PrEP uptake, adherence, and persistence remain suboptimal [8–11]. Long-acting formulations that eliminate the need for daily pills offer a promising alternative by providing sustained protection with infrequent dosing and may increase PrEP adoption among FSWs and MSWs. Long-acting injectable cabotegravir PrEP (CAB-LA) has shown superior efficacy in preventing HIV infections, with a 79% reduction in HIV acquisition in new cases observed across over 60 sites, including 20 Sub-Saharan Africa sites, among cisgender women, men who have sex with men, and transgender women who have sex with men [12–15]. By reducing the reliance on daily adherence, CAB-LA has the potential to overcome barriers that have limited the impact of oral PrEP and may be particularly well-suited to high-risk populations such as FSWs and MSWs.

The WHO endorsement of CAB-LA as a PrEP option [16] and its adoption by Uganda in 2022 [7] have generated optimism for improved HIV prevention outcomes ahead of its rollout. Nevertheless, uncertainty remains regarding the optimal service attributes and delivery models that best facilitate uptake among MSWs and FSWs. Service delivery features significantly influence an individual's decision to utilize a service [17,18]. Economic theories may help explain health consumer choices. First, Lancaster's Theory of Consumer Choice suggests that consumers derive utility not from the goods or services themselves, but from the attributes or characteristics they offer. [17, 18]. In the context of CAB-LA PrEP, service delivery characteristics may influence uptake decisions among female sex workers (FSWs) and male sex workers (MSWs). According to Lancaster's Theory of Consumer Choice, individuals derive utility not from the service itself, but from the combination of attributes offered by a particular delivery model [18]. Consequently, individuals choose one service delivery

model over another based on the combination of attributes that best aligns with their preferences. To optimize CAB-LA PrEP uptake among MSWs and FSWs, it is essential to identify the delivery attributes that facilitate uptake and to design strategies that reduce barriers while maximizing perceived utility.

The second theory, Random Utility Theory (RUT), proposes that an individual's utility from choosing a particular option consists of two components: a systematic (observable) component, determined by the attributes of the option, and a random (unobservable) component, which captures the influence of unobserved factors such as individual preferences, attitudes, or random error [19,20]. In the context of CAB-LA PrEP, an individual's decision to select a particular service delivery model is influenced by both observable attributes (e.g., location, provider gender, frequency of visits) and unobservable factors such as personal preferences and attitudes toward healthcare providers.

Research in SSA has documented preferred attributes for oral PrEP delivery among FSWs [21], including convenient dispensing locations and access to additional services, such as cervical cancer screening, at family planning clinics and NGO-run drop-in centers [21]. In Uganda, FSWs prefer receiving oral PrEP from healthcare workers at clinics, supplemented by short message service (SMS) reminders for adherence support [22]. Data on MSWs is unavailable. Other service delivery characteristics that have been documented to influence decision-making around HIV prevention service uptake include waiting time, provider gender, provider attitude, availability of medical supplies, and distance to the service location. [21,23,24]. Therefore, this study aimed to investigate the service delivery preferences for CAB-LA as an HIV PrEP option among heterosexual male and female sex workers in two urban cities in Uganda.

## Materials and methods

### Ethics statement

This study obtained ethical approval from the Infectious Diseases Institute Research Ethics Committee (IDI-REC-2024–96) and the Uganda National Council for Science and Technology (HS4747ES). All respondents provided written informed consent in English or their local language, and were compensated for their time.

### Study design and setting

Between October and December 2024, we conducted a discrete choice experiment (DCE) with MSWs and FSWs in Kampala, the capital of Uganda, and Mbarara, a city in western Uganda (combined population of 2,200,909) [25], to identify preferred service delivery attributes and models for CAB-LA PrEP. We previously established a cohort of FSWs and MSWs in Kampala and Mbarara [3,26], two cities with a high adult HIV prevalence [2,27]. DCE is a stated preference research method grounded in the economic theory of utility maximization [28].

### DCE attributes and levels

Based on existing literature, we identified several service delivery attributes that influence HIV service uptake, including service location, waiting time, provider gender, provider attitude, cost, distance to the service site, and availability of medical supplies [21,23,29–31]. However, prior studies identified location, waiting time, provider gender, and the provision of additional services as key attributes influencing preferences for PrEP service delivery [32–35]. Based on this evidence, we hypothesized that these service characteristics would similarly shape delivery preferences in this study. Prior research suggested that presenting participants with 16 hypothetical choice sets could be overly burdensome [36,37]. To reduce respondent fatigue, we aimed to include fewer choice sets. Using STATA version 17.0, we developed a statistically efficient and balanced fractional factorial DCE design comprising 10 choice cards. Each card featured four alternative service delivery options and an opt-out option, based on four key service attributes: location, waiting time, provider gender, and provision of additional services (Table 1). The opt-out option was included on each choice card to allow participants to choose neither of the options on a choice card. The choice cards were pilot-tested to ensure the clarity and language

**Table 1. DCE attributes and levels.**

| Attribute | Levels |
|---|---|
| **Dispensing location** | |
| | 1. Private pharmacy |
| | 2. Private drug shop |
| | 3. District hospital |
| | 4. Private clinic |
| | 5. MARPI clinic (dedicated clinic for most at risk populations at Mulago) |
| | 6. Community health centre |
| | 7. STI clinic |
| | 8. Family planning clinic |
| | 9. ART clinic |
| | 10. NGO-run-drop-in center |
| | 11. NGO-run-mobile outreach |
| **Provider** | |
| | 1. Male, HCW |
| | 2. Female, HCW |
| | 3. Male, peer |
| | 4. Female, peer |
| **Clinic waiting time** | |
| | 1. < 1 hour |
| | 2. 1 hour |
| | 3. 2 hours |
| **Additional service** | |
| | 1. Risk reduction counseling |
| | 2. Receive condoms |
| | 3. Cancer screening, i.e., cervical |
| | 4. Hypertension/diabetes screening |

*HCW, doctor, or nurse/midwife.*

appropriateness of the DCE question items. We pilot-tested the choice cards using a survey tool to collect demographic data to assess clarity, language appropriateness, and average completion time. The average completion time for the survey was approximately 30–40 minutes. The DCE survey questionnaire is attached as an appendix (S1 Text).

## Sampling and recruitment procedures

Study respondents were eligible if they were HIV-negative, heterosexual men or women aged 18 years or older, engaged in sex work (defined as selling sex for goods or money), and proficient in English or Luganda (local language). We estimated a sample size of 251 respondents using the rule of thumb proposed by Johnson and Orme [38–40].

$$N \geq 500\, Lmx/JS$$

where Lmx is the maximum number of levels across all attributes, J is the number of alternatives per choice set (excluding the opt-out), and S is the number of choice sets per respondent. In this study, the attribute location had the most levels (Lmax = 8), with 4 alternatives per choice set (J = 4) and 10 choice sets per respondent (S = 10). Applying the formula

yielded a minimum sample size of 100 respondents. To accommodate an anticipated 20% non-response or opt-out rate, we aimed to enroll 125 participants per site. We enrolled 124 and 127 respondents, respectively, from Mbarara and Kampala, using snowball sampling from previously identified sex work hotspots and an established cohort of MSWs and FSWs [3,26]. Sex work venues for FSWs included streets, lodges, bars/clubs, and brothels. In contrast, MSWs solicited clients through alternative channels, such as pimps, online advertising, social media platforms, and recreational venues. Recruitment in each city commenced with the purposive selection of four FSWs, each representing a venue type (streets, lodges, bars/clubs, and brothels), alongside three MSWs from our established cohort. Following their selection, they were contacted via telephone and invited to a personal meeting with a research team member at a convenient venue of their choice, where further details, including eligibility for the study and peer recruitment, were discussed. The research team comprised counselors with a minimum qualification of a Bachelor's degree, experienced in data collection, HIV counseling, and testing in sex work settings. Each participant was then given two paper coupons containing an identification number, the research team's contact information, and the study duration to distribute to potential respondents within their network. To ensure respondent safety, our research team completed ethics training before recruitment, engaged with sex worker leaders, established safety protocols (such as working in pairs on the street and sharing contact information), sought participant community input on safety, and conducted procedures at respondent-chosen venues. Data was anonymized, with no identifiers or names used, to maintain confidentiality and anonymity. FSWs and MSWs who approached a member of the research team at each study site and presented a coupon consented to undergo a rapid HIV antibody test; those testing negative for HIV consented to participate in the DCE survey. SWs diagnosed with HIV and who were not already receiving antiretroviral therapy (ART) were referred to their preferred HIV-ART clinics for further care and treatment. Upon completion of the survey, each respondent received two paper coupons to distribute to their networks until the required sample size was achieved at each study site.

## Statistical analysis

Demographic characteristics, condom use, and syphilis and HIV testing behaviors were summarized using frequencies and proportions. Pearson's Chi-square ($\chi^2$) or Fisher's exact tests were conducted to compare participants' categorical variables, and the Mann-Whitney U test was used to compare continuous variables. A mixed logit (MIXL) model with 1000 Halton simulation draws and normal distribution was employed to determine the preferred attributes for CAB-LA delivery. This approach allowed estimation of both the degree of preference and heterogeneity in preference across participants [41,42]. We used the range method to estimate conditional relative importance attributes, with higher values indicating higher importance. The differences in preference by study site (Kampala vs Mbarara), age (<25 years vs ≥ 25 years), and gender (male vs female) were also evaluated in subgroup analyses. Marginal utilities for each alternative CAB-LA delivery model were calculated using linear predictions from the final model. Finally, the preferred option among the alternative models was identified based on the highest marginal utility score. Significance was determined at 5% level. Statistical analyses were performed using R software, version 4.3.3.

## Results

### Participant characteristics

A total of 251 participants were enrolled, of whom 52 (20.7%) were MSWs. The median age was 25 years (IQR: 22–31), with no significant difference by sex (p = 0.367) (Table 2). MSWs were more likely to have higher education (26.9% vs. 4.5%, p < 0.001) and fewer children (median: 1 [IQR: 0–1] vs. 1 [0–2], p < 0.001). FSWs had been in sex work longer (24 vs. 18 months, p = 0.009), had more clients weekly (26 vs. 3, p < 0.001), and earned more (UGX 250,000 vs. 145,000, p < 0.001). Additionally, most FSWs (71.9%) worked full-time due to a lack of income alternatives, while most MSWs (57.7%) did sex work part-time (Table 2).

**Table 2. Characteristics of SWs enrolled in the DCE survey stratified by sex.**

| Characteristic | Overall, N (Col%) | Sex | | p-value |
|---|---|---|---|---|
| | | Female, N (Col%) | Male, N (Col%) | |
| *Overall* | *251* | *199 (79.3%)* | *52 (20.7%)* | |
| **Location** | | | | 0.683[C] |
| Kampala | 127 (50.6) | 102 (51.3) | 25 (48.1) | |
| Mbarara | 124 (49.4) | 97 (48.7) | 27 (51.9) | |
| **Age in completed years** | 25 (22, 31) | 25 (22, 32) | 25 (22, 29) | 0.367[M] |
| **Highest Level of Education Attained** | | | | <0.001[F] |
| Higher education | 23 (9.2) | 9 (4.5) | 14 (26.9) | |
| No education | 11 (4.4) | 9 (4.5) | 2 (3.8) | |
| Primary | 107 (42.6) | 92 (46.2) | 15 (28.8) | |
| Secondary | 110 (43.8) | 89 (44.7) | 21 (40.4) | |
| **Marital status** | | | | 0.360[F] |
| Married | 18 (7.2) | 12 (6.0) | 6 (11.5) | |
| Regular boyfriend/girlfriend | 45 (17.9) | 34 (17.1) | 11 (21.2) | |
| Single | 183 (72.9) | 149 (74.9) | 34 (65.4) | |
| Widow | 5 (2.0) | 4 (2.0) | 1 (1.9) | |
| **Number of children (biological) currently** | 1 (0, 2) | 1 (0, 2) | 1 (0, 1) | <0.001[M] |
| **Duration in sex work (In completed months)** | 24 (12, 48) | 24 (12, 48) | 18 (5, 36) | 0.009[M] |
| **Description of current work** | | | | <0.001[F] |
| Full-time, to supplement my other sources of income | 8 (3.2) | 5 (2.5) | 3 (5.8) | |
| Full time, as I have no other sources of income | 159 (63.3) | 143 (71.9) | 16 (30.8) | |
| Part-time, as I am a student | 4 (1.6) | 1 (0.5) | 3 (5.8) | |
| Part-time, as I have other sources of income | 80 (31.9) | 50 (25.1) | 30 (57.7) | |
| **Average number of clients per week** | 23 (9, 34) | 26 (20, 35) | 3 (3, 5) | <0.001[M] |
| **Average amount of money earned per week (Ugandan shillings)** | 200,000 (145,000, 350,000) | 250,000 (155,000, 400,000) | 145,000 (60,000, 250,000) | <0.001[M] |

*Col %* - *column percentage; **IQR** - interquartile range; **M**Mann-Whitney U test; **C**Pearson Chi-square; **F**Fisher's Exact test.*

## Condom use, syphilis/HIV testing behaviors, and PrEP preferences

Significant differences in condom use, syphilis/HIV testing behaviors, and PrEP use were observed among MSWs and FSWs (Table 3). Condom use at last sex was significantly lower among MSWs compared to FSWs (67.3% vs. 92.5%, p < 0.001). MSWs were also less likely to report consistent condom use with all clients (32.7% vs. 60.8%) and more likely to report non-use with regular clients (30.8% vs. 11.6%). MSWs reported slightly higher syphilis testing frequency in the past six months (median: 1 vs. 1, IQR: 1–2 vs. 0–2; p = 0.018), while HIV testing frequency and modality were similar across groups. Intention to test for HIV in the next 3 months was higher among MSWs (87.9% vs. 76.4%, p = 0.006). Awareness of PrEP was high overall but lower among MSWs (86.5% vs. 95.5%, p = 0.027). Overall, 39.4% reported current PrEP use. Among those aware of PrEP, most preferred injectable PrEP (77.3% overall), with no significant difference by sex. Willingness to use PrEP among non-users was higher among MSWs (67.3% vs. 45.7%, p = 0.001) (Table 3).

## PrEP service delivery preferences

The most valued attribute for CAB LA delivery was the dispensing location, accounting for 80% of relative importance. This was followed by the provision of additional services at the dispensing location, which accounted for 12% of relative

PLOS Global Public Health

**Table 3. Condom use, syphilis, HIV testing behaviors, and PrEP use stratified by sex.**

| Characteristic | Overall, N (Col%) | Sex | | P-value |
|---|---|---|---|---|
| | | Female, N (Col %) | Male, N (Col %) | |
| *Overall* | *251* | *199 (79.3)* | *52 (20.7)* | |
| **Used condom at last sex** | | | | <0.001ᶜ |
| No | 32 (12.7) | 15 (7.5) | 17 (32.7) | |
| Yes | 219 (87.3) | 184 (92.5) | 35 (67.3) | |
| **Description of condom use** | | | | <0.001ᶠ |
| Consistently use at all times with all clients | 138 (55.0) | 121 (60.8) | 17 (32.7) | |
| Sometimes non-use for more money | 27 (10.8) | 23 (11.6) | 4 (7.7) | |
| Sometimes non-use if I like the client by look | 7 (2.8) | 5 (2.5) | 2 (3.8) | |
| Sometimes non-use if the client requests so | 9 (3.6) | 7 (3.5) | 2 (3.8) | |
| Sometimes non-use if we test for HIV | 30 (12.0) | 20 (10.1) | 10 (19.2) | |
| Sometimes non-use with regular clients | 39 (15.5) | 23 (11.6) | 16 (30.8) | |
| Other¶ | 1 (0.4) | 0 (0) | 1 (1.9) | |
| **Syphilis testing frequency (prior 6 months) (median [IQR])** | 1 (0, 2) | 1 (0, 2) | 1 (1, 2) | 0.018ᴹ |
| **HIV testing frequency (prior months) (median [IQR])** | 2 (1, 3) | 2 (1, 3) | 2 (2, 3) | 0.18ᴹ |
| **HIV testing modality (prior 6 months)** | | | | 0.59ᶠ |
| Not tested in prior 6 months | 5 (2.0) | 5 (2.5) | 0 (0.0) | |
| Outreach | 77 (30.7) | 60 (30.2) | 17 (32.7) | |
| Self-testing | 51 (20.3) | 38 (19.1) | 13 (25.0) | |
| Visited a health facility | 118 (47.0) | 96 (48.2) | 22 (42.3) | |
| **Intention to test for HIV in the next 3 months** | | | | 0.006ᶠ |
| High intention | 181 (72.1) | 152 (76.4) | 109 (87.9) | |
| Low intention | 43 (17.1) | 29 (14.6) | 9 (7.3) | |
| No intention | 18 (7.2) | 14 (7.0) | 1 (0.8) | |
| Not sure | 9 (3.6) | 4 (2.0) | 5 (4.0) | |
| **Intention to take a serological syphilis test in the next 3 months** | | | | 0.669ᶠ |
| High intention | 136 (54.2) | 107 (53.8) | 29 (55.8) | |
| Low intention | 63 (25.1) | 48 (24.1) | 15 (28.8) | |
| No intention | 42 (16.7) | 36 (18.1) | 6 (11.5) | |
| Not sure | 10 (4.0) | 8 (4.0) | 2 (3.8) | |
| **Awareness of HIV PrEP** | | | | 0.027ᶠ |
| **Yes** | 235 (93.6) | 190 (95.5) | 45 (86.5) | |
| **Prior CAB LA awareness (N = 235)** | | | | 0.457ᶜ |
| **Yes** | 17 (7.2) | 12 (6.3) | 5 (11.1) | |
| **Currently using PrEP use (N = 235)** | | | | <0.54ᶜ |
| **Yes, oral** | 99 (39.4) | 80 (42.1) | 19 (42.2) | |
| | | | | 0.614ᶠ |
| **Preferred PrEP option** | | | | |
| **Daily oral** | 44 (17.5) | 34 (17.1) | 10 (19.2) | |
| **Injectable** | 194 (77.3) | 156 (78.4) | 38 (73.1) | |
| **Vaginal ring** | 4 (1.6) | 3 (1.5) | 1 (1.9) | |
| **On-demand** | 9 (3.6) | 6 (3.0) | 3 (5.8) | |
| **Willingness to use PrEP (none-current user), N = 152** | | | | 0.001ᶜ |
| **Very high or high** | 126 (82.9%) | 54 (45.7) | 22 (67.3%) | |

*Col % - column percentage; **IQR** - interquartile range; **M** Mann-Whitney U test; Pearson Chi-square; **F**Fisher's Exact test. ¶-She tested me, and we agreed not to use condoms.*

importance. In contrast, provider gender or cadre (4.8%) and waiting time (3.2%) were considered the least important attributes (Table 4). In subgroup analysis by gender, dispensing location (relative importance 81.1% MSWs, 82.9% FSWs) and additional services (10.8% MSWs, 9% FSWs) remained the most ranked delivery attributes (Table 5). Based on utility scores for various attribute combinations, the most preferred CAB LA delivery model consisted of: delivery at a private clinic, waiting time for no more than an hour, service provided by a peer, and availability of psychosocial support services (Table 6). The top-ranked CAB LA delivery models, as determined by preference weights, featured a private pharmacy/clinic as the delivery location and included additional services such as psychosocial support, prostate or cervical cancer screening, and risk reduction counseling.

**Table 4. Results of random parameter logit model (mixed logit model) for the CAB LA model delivery attributes*.**

| Attributes and levels[a] | Coefficients | 95% CI | P-value† | SD | 95% CI | P-value | Relative importance¶ |
|---|---|---|---|---|---|---|---|
| Constant | -2.55 | -5.01- -0.08 | 0.04 | – | – | – | – |
| **Dispensing location** | | | | | | | **80.0%** |
| *Private drug shops* | *Reference* | | | | | | |
| ART clinic | 9.92 | 3.55 - 16.29 | <0.01 | 0.03 | -3.52 - 3.53 | 0.99 | |
| Private clinic | 36.99 | <-50 ->50‡ | 0.99 | 0.0003 | -11.66-11.66 | 0.99 | |
| NGO-run drop-in center | 11.62 | 1.94 - 21.30 | 0.02 | 0.02 | -2.55 - 2.55 | 0.99 | |
| Private pharmacy | 34.48 | <-50 ->50‡ | 0.99 | 0.0002 | -11.72 -11.72 | 0.99 | |
| Family planning clinic | 11.25 | 2.40 - 20.10 | 0.01 | 0.02 | -3.74 - 3.75 | 0.99 | |
| District hospital | 7.21 | -0.64 -15.06 | 0.07 | 0.01 | -3.68 - 3.68 | 0.99 | |
| STI clinic | 6.70 | 2.00 - 11.42 | 0.01 | 0.02 | -3.38 - 3.40 | 0.99 | |
| Community health center | 7.71 | 0.92 - 14.51 | 0.03 | 0.007 | -3.66 - 3.68 | 0.99 | |
| MARPs clinic | 10.32 | 0.93 - 19.70 | 0.03 | 0.003 | -2.43 - 2.44 | 0.99 | |
| NGO-run mobile outreach | 7.67 | -2.42 - 7.77 | 0.14 | 0.01 | -2.54 - 2.54 | 0.99 | |
| **Clinic waiting time** | | | | | | | **3.2%** |
| < 1 hour | -0.07 | -6.53 - 6.38 | 0.98 | 0.02 | -2.25 - 2.25 | 0.99 | |
| 1 hour | -1.71 | -6.63 - 3.21 | 0.50 | 0.04 | -2.28 - 2.33 | 0.98 | |
| *2 hours* | *Reference* | | | | | | |
| **Provider** | | | | | | | **4.8%** |
| Male, HCW | Reference | | | | | | |
| Female, HCW | -1.67 | -9.40 - 6.07 | 0.67 | 0.007 | -2.08 - 2.10 | 0.99 | |
| Male, peer | -0.33 | -5.66 - 5.01 | 0.91 | 0.002 | -2.26 - 2.31 | 0.98 | |
| Female, peer | -2.54 | -8.57 - 3.49 | 0.41 | 0.009 | -2.36 - 2.43 | 0.98 | |
| **Additional services at the site** | | | | | | | **12.0%** |
| *Receiving contraceptive services, e.g., condoms* | *Reference* | | | | | | |
| Psychosocial support | 6.30 | -0.33 -12.93 | 0.06 | 0.0009 | -3.24 - 3.24 | 0.99 | |
| Check for hypertension and diabetes | 5.01 | 3.09 - 6.94 | <0.01 | 0.02 | -2.42 - 2.52 | 0.97 | |
| Check for cancer, i.e., cervical or prostate | 5.34 | 0.12 - 10.56 | 0.04 | 0.001 | -2.59 - 2.59 | 0.99 | |
| Receive risk counseling | 1.18 | -2.41 - 4.77 | 0.52 | 0.04 | -2.44 - 2.50 | 0.98 | |

*CI - Confidence interval; SD - standard deviation for preference heterogeneity (random component of the model coefficients); † P value testing the hypothesis that standard deviation (heterogeneity across individuals' preferences) equals '0' number of observations = 12,520; ‡ the confidence intervals for private clinic and private pharmacy attribute levels were too wide, thus used cut-off -/+ 50. This could be due to multicollinearity effect induced by having both private clinic and private pharmacy attributes within the model.*

 

**Table 5. Results of random parameter logit model (mixed logit model) for the CAB LA PrREP model attributes* by sex.**

| Attributes and levels[a] | MALES | | | FEMALES | | |
|---|---|---|---|---|---|---|
| | Coefficients | SD | Relative importance¶ | Coefficients | SD | Relative importance¶ |
| Constant | -1.33 | | | -1.26 | – | |
| **Dispensing location** | | | 81.1% | | | 82.9% |
| *Private drug shops* | Reference | | | Reference | | |
| ART clinic | 7.05 | <0.01 | | 6.77 | 0.11 | |
| Private clinic | 31.72 | <0.01 | | 40.69 | <0.01 | |
| NGO-run drop-in center | 8.20 | <0.01 | | 7.79 | 0.03 | |
| Private pharmacy | 29.40 | <0.01 | | 38.29 | 0.01 | |
| Family planning clinic | 7.91 | 0.01 | | 7.78 | 0.13 | |
| District hospital | 4.67 | <0.01 | | 4.37 | 0.11 | |
| STI clinic | 5.23 | 0.01 | | 5.33 | 0.53 | |
| Community health center | 5.14 | 0.01 | | 4.90 | 0.34 | |
| MARPs clinic | 6.95 | 0.01 | | 7.44 | 1.28 | |
| NGO-run mobile outreach | 5.17 | <0.01 | | 4.87 | 0.25 | |
| **Clinic waiting time** | | | 3.9% | | | 4.0% |
| < 1 hour | 1.04 | <0.01 | | 1.42 | 0.26 | |
| 1 hour | -0.71 | 0.03 | | -0.56 | 0.07 | |
| *2 hours* | Reference | | | Reference | Reference | |
| **Provider** | | | 4.2% | | | 4.0% |
| Male, HCW | Reference | | | Reference | Reference | |
| Female, HCW | -0.96 | 0.01 | | -0.85 | 0.27 | |
| Male, peer | -0.16 | 0.02 | | -0.02 | 0.04 | |
| Female, peer | -1.88 | 0.03 | | -1.97 | 0.15 | |
| **Additional services at the site** | | | 10.8% | | | 9.0% |
| *Receiving contraceptive services, e.g., condoms* | Reference | | | Reference | Reference | |
| Psychosocial support | 4.83 | <0.01 | | 4.85 | 0.25 | |
| Check for hypertension and diabetes | 3.79 | 0.05 | | 4.03 | 0.67 | |
| Check for cancer, i.e., cervical or prostate | 4.26 | <0.01 | | 4.44 | 0.75 | |
| Receive risk counseling | 1.15 | 0.03 | | 1.29 | 0.43 | |

[a]Dummy coded attributes (coefficient of the reference category is constrained to be 0) SD standard deviation for preference heterogeneity (random component of the model coefficients), † P value testing the hypothesis that standard deviation (heterogeneity across individuals' preferences) equals '0'Number of observations = 12,520, ‡ The confidence intervals for private clinic and private pharmacy attribute levels were too wide, thus used cut-off -/ +50. This could be due to multicollinearity effect induced by having both private clinic and private pharmacy attributes within the model.

## Discussion

In this DCE exploring CAB-LA PrEP delivery preferences among MSWs and FSWs in Uganda, most respondents preferred CAB-LA over other PrEP options. Among non-current PrEP users, willingness to initiate PrEP was higher among MSWs than FSWs. However, both FSWs and MSWs prioritized receiving CAB-LA from private clinics or pharmacies that provide additional services such as psychosocial support, risk reduction counseling, and cancer screening. Additionally, the top three preferred CAB-LA delivery models emphasized access through private facilities with short waiting times, female or peer providers, and integrated health services. The most preferred model was a private clinic with a waiting time of less than one hour, attended by a female peer provider, and offering psychosocial support. This was followed by a private pharmacy providing cancer screening, staffed by a female health worker, also with a waiting time of under one hour. The third-ranked model involved a private pharmacy with a female peer provider, provision of contraceptives or condoms,

**Table 6. Hypothetical CAB LA delivery models ranks and utility score\*.**

| Model | Choice card no | Alternatives | Utility score | Rank |
|---|---|---|---|---|
| Private clinic/<1hr/female peer/receive psychosocial support | 10 | 2 | 43.30 | 1 |
| Private pharmacy/1hr/female HCW/check for cancer, i.e., cervical or prostate | 10 | 4 | 39.08 | 2 |
| Private pharmacy/2hrs/female peer/contraceptives, e.g., condom | 2 | 1 | 34.57 | 3 |
| FP clinic/2hr/female HCW/check for diabetes and hypertension | 1 | 1 | 12.05 | 4 |
| District hospital/2hrs/female HCW/receive psychosocial support | 8 | 1 | 9.29 | 5 |
| Mobile outreach/<1hr/female HCW/check for cancer, i.e., cervical or prostate | 7 | 2 | 8.73 | 6 |
| FP clinic/<1hr/female peer/ receive risk counselling | 3 | 1 | 7.34 | 7 |
| ART clinic/1hr/male HCW/receive risk counselling | 8 | 2 | 6.84 | 8 |
| Community health center/1hr/female HCW/check for diabetes and hypertension | 3 | 2 | 6.80 | 9 |
| District hospital/<1hr/male HCW/ contraceptives, e.g., condom | 9 | 4 | 4.59 | 10 |
| Private clinic/<1hr/male HCW/check for cancer, i.e., cervical or prostate | 1 | 2 | 2.55 | 11 |
| STI clinic/2hr/male HCW/ contraceptives, e.g., condom | 4 | 4 | 2.55 | 11 |
| STI clinic/1hr/male HCW/check for diabetes and hypertension | 7 | 4 | 2.55 | 11 |
| ART clinic/<1hr/female HCW/check for diabetes and hypertension | 6 | 2 | 2.55 | 11 |
| Mobile outreach/2hrs/male HCW/receive psychosocial support | 3 | 4 | 2.55 | 11 |
| Community health center/<1hr/male HCW/ contraceptives, e.g., condom | 5 | 4 | 2.55 | 11 |
| STI clinic/<1hr/male peer/receive risk counselling | 2 | 4 | 2.55 | 11 |
| MARPI clinic/2hrs/female HCW/receive risk counselling | 4 | 1 | 2.55 | 11 |
| Mobile outreach/<1hr/female peer/check for diabetes and hypertension | 8 | 4 | 2.55 | 11 |
| Drop-in center/1hr/male peer/ contraceptives, e.g., condom | 10 | 1 | 2.55 | 11 |
| Community health center/2hrs/male peer/check for cancer, i.e., cervical or prostate | 9 | 1 | 2.55 | 11 |
| MARPI clinic/1hr/female peer/check for diabetes and hypertension | 9 | 2 | 2.55 | 11 |
| Private clinic/2hrs/male peer/check for diabetes and hypertension | 5 | 1 | 2.55 | 11 |
| Private pharmacy/ <1hr/male peer/receive psychosocial support | 6 | 1 | 2.55 | 11 |
| FP clinic/1hr/male HCW/receive psychosocial support | 2 | 2 | 2.55 | 11 |
| MARPI clinic/2hrs/female peer/check for cancer, i.e., cervical or prostate | 6 | 4 | 2.55 | 11 |
| Drop-in center/<1hr/female HCW/receive psychosocial support | 4 | 2 | 2.55 | 11 |
| District hospital/1hr/female peer/check for cancer, i.e., cervical or prostate | 5 | 2 | 2.55 | 11 |
| ART clinic/2hrs/male peer/ contraceptives, e.g., condom | 7 | 1 | 2.55 | 11 |
| Mobile outreach/1hr/male peer/receive risk counselling | 1 | 4 | 2.55 | 11 |

and a waiting time under two hours. Ranked fourth was a family planning clinic offering screening for diabetes and hypertension, attended by a female provider, with a waiting time under two hours.

The strong preference for CAB-LA observed in this study suggests that introducing injectable PrEP options could increase PrEP acceptability and uptake among SWs in Uganda. Expanding the variety of PrEP modalities has been shown to improve user choice, acceptability, and adherence [43–45]. CAB-LA is administered every two months following an initial loading dose, and this relatively infrequent schedule likely contributes to its preference over daily oral PrEP by reducing the adherence burden and supporting sustained protection. Emerging options such as injectable lenacapavir, administered only twice a year [46,47], may further enhance acceptability among SWs by providing an even more convenient and less frequent dosing option. Although limited data exist on PrEP preferences among MSWs, similar preferences for injectable PrEP options have been reported globally among FSWs, MSM, and other key populations [48–54]. Studies conducted in SSA—specifically in Kenya, Tanzania, South Africa, and Malawi—among FSWs and vulnerable youth report high preference rates (74–88%) for injectable PrEP [21,55–57]. While we did not explore reasons for this preference, prior

studies suggest that ease, convenience, perceived effectiveness, and privacy make injectable PrEP appealing to SWs [53,58,59]. Our finding that willingness to initiate PrEP was higher among MSWs than FSWs may reflect greater perceived HIV risk. In this study, MSWs reported lower rates of condom use and consistency compared to FSWs, consistent with our previous findings [3].

Consistent with prior SSA studies on oral PrEP, dispensing location, and additional services were highly valued by FSWs and adolescent girls [60–62]). In our study, both MSWs and FSWs preferred to access CAB-LA PrEP from private pharmacies and private clinics offering additional services, highlighting private sector channels as an alternative strategy for scaling up injectable PrEP services. The preference for CAB-LA delivery through private facilities with short waiting times, female or peer providers, and integrated services highlights key priorities for effective rollout among SWs [31,63]. Engaging private pharmacies, ensuring client-centered and stigma-free care, and offering integrated health services [64,65] may improve acceptability and uptake. An integrated service delivery model known as MARPI—a center of excellence for key population services—is widely recognized in Uganda [66]. Although we anticipated that both male and female sex workers would prefer this model for CAB-LA PrEP delivery, our analysis revealed that MARPI was ranked lower (utility score: 2.55) compared to private clinics (utility score: 43.3) or private pharmacies (utility score: 39.8). This suggests a possible preference for more discreet service delivery settings, where individuals may feel less exposed or less likely to be identified as sex workers [67]. Notably, the private pharmacy model has already been scaled up successfully in Kampala for ART delivery [68], consistent with prior evidence on the acceptability of private pharmacies for HIV prevention services in SSA [29]. Community pharmacies in the US have successfully implemented PrEP programs, with a notable example being a Seattle study where CAB-LA PrEP delivery achieved 94% on-time injection adherence and 81% retention in care at 26 weeks [69,70]. Our study, however, highlights the importance of integrating additional services, such as psychosocial support, risk reduction counseling, and cancer screening, into pharmacy-based CAB-LA PrEP delivery models, similar to Malawi, where FSWs expressed a desire for oral PrEP services bundled with cervical cancer screening at NGO-run clinics [21]. In Uganda, private pharmacies play a crucial role in distributing products for preventing malaria, tuberculosis, and HIV. They provide artemisinin-based combination therapies, malaria rapid diagnostic kits, HIV self-test kits, PrEP, condoms, emergency contraceptives, and ART refills [68,71–75]. Nonetheless, pharmacy-based delivery models may pose financial barriers, especially for SWs seeking integrated services or those with limited income, unless costs are subsidized. Our findings highlight the importance of understanding user preferences for dosing schedules, delivery models, and supportive services, not only for CAB-LA but also for emerging six-monthly injectables such as lenacapavir. Implementation research and costing studies are needed to ensure feasibility, affordability, and sustainability. Additionally, public sector services remain a key platform for HIV prevention. They may require adaptation to improve efficiency, provider attitudes, and integration of services [64,65] to better meet the needs of SWs. CAB-LA offers a promising opportunity to expand HIV prevention among SWs in Uganda, while emerging six-monthly injectables such as lenacapavir hold future potential to further enhance convenience, acceptability, and adherence. To maximize impact, implementation strategies must prioritize sustainable, integrated, and equitable delivery models that address user preferences, service accessibility, and adherence support for FSWs and MSWs.

The present study's strengths include being the first to investigate preferences for CAB-LA PrEP delivery among MSWs and FSWs in Uganda and using a DCE to provide insights/delivery strategies that can inform future implementation. However, limitations exist: the sample from two cities may not represent all Ugandan MSWs and FSWs, especially those living in more rural settings. Results may also not apply to other African settings; in Malawi, FSWs preferred family planning clinics or NGO-run drop-in centers for oral PrEP [60]. DCEs' hypothetical scenarios may not mirror real-world choices. Additionally, we did not explore the reasons why SWs in this cohort preferred private pharmacies and clinics. Furthermore, we did not assess the role of stigma, harassment, or gender-based violence—contextual factors that could influence PrEP preferences. Despite these limitations, our findings align with studies on oral PrEP delivery preferences [21,76].

## Conclusions

In this study, willingness to initiate PrEP was high among MSWs. Expanding injectable PrEP through private sector channels could improve access among sex workers. However, implementation research is needed to guide integration of supportive services and assess willingness to pay for pharmacy- or clinic-based delivery.

## Supporting information

**S1 Text. Full survey questionnaire and discrete choice experiment materials provided to participants.**
(DOCX)

**S1 Data. De-identified participant-level dataset used for analysis (demographic variables).**
(DTA)

**S2 Data. De-identified participant-level dataset used for analysis (DCE items).**
(DTA)

## Acknowledgments

We extend our sincere gratitude to the leadership of the cities of Kampala and Mbarara for their support and facilitation of the data collection process. Additionally, we acknowledge the contributions of Robert Zimula and Jolly Twinomugisha, who served as research assistants.

## Author contributions

**Conceptualization:** Richard Muhindo.

**Data curation:** Richard Muhindo, Flavia Dhikusooka, Joseph Musaazi.

**Formal analysis:** Andrew Mujugira, Whitney Irie, Flavia Dhikusooka, Joseph Musaazi.

**Methodology:** Rachel King, Andrew Mujugira, Whitney Irie, Patience Muwanguzi, Barbara Castelnuovo.

**Project administration:** Richard Muhindo.

**Supervision:** Rachel King, Barbara Castelnuovo.

**Validation:** Rachel King, Andrew Mujugira.

**Writing – original draft:** Richard Muhindo.

**Writing – review & editing:** Rachel King, Andrew Mujugira, Whitney Irie, Patience Muwanguzi, Flavia Dhikusooka, Joseph Musaazi, Barbara Castelnuovo.

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
