## [Decision Letter · Decision Letter 0]

10 Jul 2025

PGPH-D-25-01202

Preferences for Long-Acting Injectable HIV Pre-exposure Prophylaxis Service Delivery Among Male and Female Sex Workers in Uganda: A Discrete Choice Experiment

Dear Dr. Muhindo,

Thank you for submitting your manuscript to PLOS Global Public Health. After careful consideration, we feel that it has merit but does not fully meet PLOS Global Public Health’s publication criteria as it currently stands. Therefore, we invite you to submit a revised version of the manuscript that addresses the points raised during the review process.

The manuscript has been assessed by three reviewers and their comments are available below. Whilst all reviewers noted the value of the current research they have requested revisions to the manuscript to improve the clarity and detail of the methodology and results sections; and depth of discussion. Please review their comments and make the appropriate revisions to address their concerns. 

We look forward to receiving your revised manuscript.

Kind regards,

Emma Campbell, Ph.D

Staff Editor

Journal Requirements:

Reviewers' comments:

Reviewer's Responses to Questions

**Comments to the Author**

1. Does this manuscript meet PLOS Global Public Health’s publication criteria?

Reviewer #1: Yes

Reviewer #2: Yes

Reviewer #3: Yes

2. Has the statistical analysis been performed appropriately and rigorously?

Reviewer #1: Yes

Reviewer #2: Yes

Reviewer #3: Yes

3. Have the authors made all data underlying the findings in their manuscript fully available (please refer to the Data Availability Statement at the start of the manuscript PDF file)?

Reviewer #1: Yes

Reviewer #2: Yes

Reviewer #3: Yes

4. Is the manuscript presented in an intelligible fashion and written in standard English?

Reviewer #1: Yes

Reviewer #2: Yes

Reviewer #3: Yes

Reviewer #1: This is a good paper that brings out a critical component of preferences by location, waiting time, provider gender, and provision of additional services. Considering that the world is evolving and newer products like Lenacapavir are being introduced and will primarily rely on the established PrEP programs.

I have a few follow-up questions and clarifications to the author

1. The study does not specify prior use, but the population likely included both PrEP-experienced and PrEP-naïve individuals. This could affect how preferences were formed. Can the author clarify if this information is available?

2. The study was conducted in urban settings; there was no representation from rural areas, which may limit the generalizability to rural sex worker populations and bearing in mind that most PrEP services are in urban settings. This could be highlighted as a study limitation.

3. While participants varied in age and duration in sex work, the study did not explicitly report on gender identity diversity, yet it is a critical point mentioned in this study for determining preferences (e.g., transgender SWs) or levels of mobility and HIV risk, which would be important to determine preferences.

4. While the study did not deeply explore the influence of stigma, harassment, or gendered violence on preferences, these remain critical contextual factors for both access and uptake of services and should be integrated in future research. The preference of services is also determined by access/availability, which also determines uptake

5.While Private clinics and pharmacies that offer additional health services were the most preferred delivery models for CAB-LA PrEP among both groups. Uganda prides in having a very successful integrated service delivery model called MARPI, which are KP services integrated in the health sector. Did the authors *include such in their questionnaire? This is becoming increasingly important in the current funding landscape, where KP services remain under-prioritized amid a balance between lifesaving and prevention narratives.

6. The author highlights the private sector and others as Feasible Options for delivering CAB LA and leveraging private clinics and pharmacies aligns with the study's findings and may be feasible if proper training and regulation are ensured. It would be great for the author to highlight if similar arrangements are available in the country to roll out other public health interventions through the private sector. E.g. Self self-testing, private condom dispensing, Family planning or even Malaria treatment and other services to strengthen the findings.

7. The study used trained interviewers and implemented informed consent protocols to ensure ethical participation. However, the methods section does not detail protections from harm, and ethical safeguards which were likely approved by local ethics boards given the vulnerable population involved.

8.I find it suprising for the findings that give minimal importance to provider gender and wait time challenges which are common assumptions and underscores that accessibility and comprehensiveness matter more to this. population.Could the author confirm this ?

Reviewer #2: This study aimed to understand the preferences for long-acting injectable cabotegravir (CAB-LA) among HIV-negative male and female sex workers (MSWs and FSWs) in Uganda using a discrete choice experiment (DCE). The findings showed MSWs and FSWs both prioritized dispensing location and provision of additional services. This study has some merits, but several concerns needed to be addressed before publication.

1. Abstract: The Results section does not seem to present all the important findings of the study, including the preferred delivery model. In addition, the final sentence appears to be a repetition of the previous one; please make it more concise. Based on the current results, it is difficult to conclude that 'introducing injectable PrEP in Uganda could boost acceptability and uptake among sex workers'. Please clarify this.

2. Methods: The literature review revealed several key attributes influencing decision-making regarding HIV prevention service uptake. However, the authors finally only included four of them without any clear explanations. Please clarify this. In addition, the authors mentioned 16 hypothetical choice sets, but you didn’t clarify how to get these 16 choice sets. Please provide the formula of the rule of thumb and clarify how to get the sample size.

3. Discussion: The authors only discussed the most preferred attribute, failing to address other findings, such as the preferred delivery model. In addition, the results should be compared to evidence from other parts of the world, not just SSA.

4. Conclusions: It is difficult to conclude the first sentence base on the current results or the discussion.

Reviewer #3: Thank you for the opportunity to review the submission PGPH-D-25-01202 entitled “Preferences for Long-Acting Injectable HIV Pre-exposure Prophylaxis Service Delivery Among Male and Female Sex Workers in Uganda: A Discrete Choice Experiment.” This paper analyzes a sample of 251 sex workers regarding long-acting injectable cabotegravir (CAB-LA) using a discrete choice experiment (DCE).

Overall, I think this is important work that contributes significantly to the existing literature. I do have some suggestions for revisions that I think would enhance impact of the manuscript.

Background:

1. Authors establish that sex workers are an important group in HIV prevention efforts and that CAB-LA has the potential for preventing HIV in this population. They highlight the lack of data on male sex workers (MSWs). While data exists for preferences for daily oral PrEP delivery, there is less regarding CAB-LA. This sufficiently establishes the significance of the current study.

Method:

2. I think the discussion of underlying theories of choice may be moved into the background.

3. The team’s previous literature review regarding service preferences may also be useful information in the background.

Results:

4. Differences are largely discussed between sites in the bivariates. Given the emphasis on MSWs, I wonder if it would be useful to report any significant differences in bivariate associations between MSWs and FSWs.

5. I am assuming it’s not possible to examine the differences between MSW and FSW preferences in the model. (I am not as familiar with DCE). If it is possible, I would suggest commenting on this as it is sold as a strength of the study.

Discussion:

6. I would suggest discussing MSW a bit more in the discussion, as this is a novel aspect of the study that is emphasized in the background.

7. It may be worth discussing the implications of a private vs. public pharmacy in the context of Uganda. This is the preference, but does it have implications like a higher fee? Are private pharmacies as available as public pharmacies (e.g. equivalently distributed)?

8. The findings point to integrated clinical settings and the authors point to a similar model in Malawi. Do such clinics exist in Uganda? I think more comment on the current feasibility or infrastructure in Uganda would be helpful context for this finding.

Overall:

9. I think this is an important study that contributes important information to the field.

10. I would recommend emphasizing the MSWs in the results and discussion more, as this is a novel component of the study that is emphasized in the background section. As currently presented I feel a bit “let down” as a reader, because I was expecting more commentary on the MSWs based on the background section as presented.

**Do you want your identity to be public for this peer review?** For information about this choice, including consent withdrawal, please see our Privacy Policy

Reviewer #1: **Yes: ** Helgar Musyoki

Reviewer #2: No

Reviewer #3: No

---

## [Decision Letter · Decision Letter 1]

30 Aug 2025

PGPH-D-25-01202R1

Preferences for long-acting injectable HIV pre-exposure prophylaxis service delivery among male and female sex workers in Uganda: A discrete choice experiment

Dear Dr. Muhindo,

Thank you for submitting your manuscript to PLOS Global Public Health. After careful consideration, we feel that it has merit but does not fully meet PLOS Global Public Health’s publication criteria as it currently stands. Therefore, we invite you to submit a revised version of the manuscript that addresses the points raised during the review process.

The manuscript has been evaluated by two reviewers, and their comments are available below. Could you please revise the manuscript to carefully address the concerns raised?

We look forward to receiving your revised manuscript.

Kind regards,

Johanna Pruller, Ph.D.

PLOS Staff Editor

Journal Requirements:

Reviewers' comments:

Reviewer's Responses to Questions

**Comments to the Author**

Reviewer #1: All comments have been addressed

Reviewer #2: All comments have been addressed

publication criteria?

Reviewer #1: Yes

Reviewer #2: Yes

3. Has the statistical analysis been performed appropriately and rigorously?

Reviewer #1: Yes

Reviewer #2: I don't know

4. Have the authors made all data underlying the findings in their manuscript fully available (please refer to the Data Availability Statement at the start of the manuscript PDF file)?

Reviewer #1: Yes

Reviewer #2: No

5. Is the manuscript presented in an intelligible fashion and written in standard English?

Reviewer #1: Yes

Reviewer #2: Yes

Reviewer #1: The paper reads very well and is relevant to the current public health body of knowledge.

The study is also timely, novel, and policy-relevant, addressing injectable CAB-LA PrEP among sex workers in Uganda.

The introduction could be strengthened by linking the HIV burden among sex workers with the promise of CAB-LA.

In the methods section, it would be useful to explain why specific attributes such as location, waiting time, provider gender, and extra services were chosen for the discrete choice experiment.

Further, the results would flow more clearly if they were organized into subsections, moving from participant characteristics to formulation preference and then to service delivery preferences.

An important omission is the frequency of CAB-LA, which should be explicitly stated as dosing every two months after an initial lead-in. In the interpretation, highlighting that the reduced dosing frequency helps explain why injectable PrEP is favored over oral PrEP would strengthen the discussion.

It would also be valuable to acknowledge the equity concern that private pharmacy or clinic delivery could exclude lower-income sex workers unless subsidized.

From a language and style perspective, shortening long sentences, reducing repetition of percentages, and improving readability would make the paper more polished.

It is my opinion that the author adds some looking forward context about lenacapavir, a six-monthly injectable, in recommendations, probably to position the findings as forward-looking and relevant beyond CAB-LA.

Finally, the paper would benefit from a stronger closing statement emphasizing CAB-LA’s potential now and lenacapavir’s future promise, while stressing the need for sustainable, integrated, and equitable delivery models. This is to improve the relevance of this paper and make it forward-looking

Reviewer #2: All of my comments have been addressed. One last minor comment is that, while the participants were recruited from previously identified sex work hotspots and an established cohort of MSWs and FSWs, they were not all recruited from the established cohort. Therefore, I suggest that the author not use "cohort" to describe the participants in this study.

**Do you want your identity to be public for this peer review?** For information about this choice, including consent withdrawal, please see our Privacy Policy

Reviewer #1: **Yes: ** Helgar Musyoki

Reviewer #2: No

---

## [Decision Letter · Decision Letter 2]

18 Sep 2025

Preferences for long-acting injectable HIV pre-exposure prophylaxis service delivery among male and female sex workers in Uganda: A discrete choice experiment

PGPH-D-25-01202R2

Dear Dr. Muhindo,

We are pleased to inform you that your manuscript 'Preferences for long-acting injectable HIV pre-exposure prophylaxis service delivery among male and female sex workers in Uganda: A discrete choice experiment' has been provisionally accepted for publication in PLOS Global Public Health.

Best regards,

Julia Robinson

Executive Editor

Reviewer #1:

Reviewer Comments (if any, and for reference):

Reviewer's Responses to Questions

**Comments to the Author**

Reviewer #1: All comments have been addressed

publication criteria?

Reviewer #1: Yes

3. Has the statistical analysis been performed appropriately and rigorously?

Reviewer #1: Yes

4. Have the authors made all data underlying the findings in their manuscript fully available (please refer to the Data Availability Statement at the start of the manuscript PDF file)?

Reviewer #1: Yes

5. Is the manuscript presented in an intelligible fashion and written in standard English?

Reviewer #1: Yes

Reviewer #1: No furthers comments

**Do you want your identity to be public for this peer review?** For information about this choice, including consent withdrawal, please see our Privacy Policy

Reviewer #1: **Yes: ** Helgar Musyoki
